# Genetic strategies for sex-biased persistence of gut microbes across human life

Chiara Tarracchini[1], Giulia Alessandri[1], Federico Fontana[1,2], Sonia Mirjam Rizzo[1], Gabriele Andrea Lugli[1], Massimiliano Giovanni Bianchi[3,4], Leonardo Mancabelli[3,4], Giulia Longhi[1], Chiara Argentini[1], Laura Maria Vergna[1], Rosaria Anzalone[2], Alice Viappiani[2], Francesca Turroni[1,4], Giuseppe Taurino [3,4], Martina Chiu[3], Silvia Arboleya[5], Miguel Gueimonde [5], Ovidio Bussolati [3,4], Douwe van Sinderen [6], Christian Milani [1,4,7] ✉ & Marco Ventura [1,4,7] ✉

Although compositional variation in the gut microbiome during human development has been extensively investigated, strain-resolved dynamic changes remain to be fully uncovered. In the current study, shotgun metagenomic sequencing data of 12,415 fecal microbiomes from healthy individuals are employed for strain-level tracking of gut microbiota members to elucidate its evolving biodiversity across the human life span. This detailed longitudinal meta-analysis reveals host sex-related persistence of strains belonging to common, maternally-inherited species, such as *Bifidobacterium bifidum* and *Bifidobacterium longum* subsp. *longum*. Comparative genome analyses, coupled with experiments including intimate interaction between microbes and human intestinal cells, show that specific bacterial glycosyl hydrolases related to host-glycan metabolism may contribute to more efficient colonization in females compared to males. These findings point to an intriguing ancient sex-specific host-microbe coevolution driving the selective persistence in women of key microbial taxa that may be vertically passed on to the next generation.

The intestinal microbial community of an infant gradually assembles through a patterned developmental process following birth. In particular, the first three years of life represent a critical window of opportunity during which short-term changes in the gut microbiota composition occur in conjunction with the rapid physical development of the newborn[1,2]. This initial dynamic process eventually evolves towards stable microbe-host interactions that are of paramount importance to impart beneficial effects on host health, such as metabolism of non-digestible dietary carbohydrates and stimulation of endogenous intestinal mucus production, vitamin synthesis, development and homeostasis of the immune system, as well as protection against pathogens[3]. In particular, it is known that, at species level, the establishment of a stable gut microbiota occurs through two main diet-guided stages in early childhood[4,5], with the first one taking place immediately after birth when exclusive milk-feeding begins[6,7]. This stage is characterized by a gut microbial community which, upon being partly (vertically) transmitted from the mother and partly acquired through contact with the surrounding environment during and after delivery[8–10], represent the first microbial colonizers of the infant gut due to their ability to directly or indirectly metabolize human milk oligosaccharides (HMOs)[11,12]. Another transition occurs during the weaning period, typically around the age of six months, when infants are gradually introduced to a solid and more varied diet, which offers new ecological niche colonization opportunities[13–15].

[1]Laboratory of Probiogenomics, Department of Chemistry, Life Sciences, and Environmental Sustainability, University of Parma, Parma, Italy. [2]GenProbio srl, Parma, Italy. [3]Department of Medicine and Surgery, University of Parma, Parma, Italy. [4]Interdepartmental Research Centre "Microbiome Research Hub", University of Parma, Parma, Italy. [5]Department of Microbiology and Biochemistry of Dairy Products, Instituto de Productos Lácteos de Asturias, CSIC, 33300 Villaviciosa, Spain. [6]APC Microbiome Institute and School of Microbiology, Bioscience Institute, National University of Ireland, T12YT20 Cork, Ireland. [7]These authors contributed equally: Christian Milani, Marco Ventura. ✉e-mail: christian.milani@unipr.it; marco.ventura@unipr.it

Whole-metagenome shotgun (WMGS) sequencing represents a powerful tool to disentangle the composition of complex microbial communities and retrieve genomes belonging to hard-to-culture microbial species[16–18]. Accordingly, several recent longitudinal studies have investigated the infant gut microbiome composition, highlighting the sequential age-related changes at species level[19,20]. For example, bifidobacterial species, key microbial taxa of the infant gut microbiota, can persist at a lower level (2–14% relative abundance) throughout adulthood and can subsequently be passed on to the next host generation by mother-to-infant vertical transmission[21,22]. Nevertheless, the impact of the host biological sex on the assembly and maintenance of the gut microbial community remains poorly investigated.

In the current study, a total of 400 longitudinal fecal metagenomes from 124 healthy infants (0–3 years) were investigated to inspect the intra-species variations underlying the assembly of the infant gut microbiome during the first two years following birth. In this context, we assessed if particular gut microbiota members elicited higher persistence in female infants (compared to male counterparts), perhaps to maintain cross-generational transmission. These analyses, coupled with inspection of shotgun metagenomic data from 12,415 healthy subjects (6545 females and 5870 males) aged from a few days to 90 years, allowed the identification of two glycosyl hydrolases, i.e., members of GH101 and GH136, which appear associated with persistence of *B. bifidum* and *B. longum* strains with preferential presence in the female gastrointestinal tract. Moreover, sex-related (bifido)bacterial resilience was validated in vivo by retrospective human clinical trial data involving the supplementation of bifidobacterial strains displaying a persistent vs. non-persistent genotype.

## Results

### Strain dynamics of the gut-associated microbiota within the first 24 months of life

Shotgun metagenomics sequencing approaches were applied to the microbiomes of 11 healthy, vaginally delivered, full-term (>37 weeks of gestation) newborns, which were longitudinally sampled at 1-, 6-, 12-, and 24-months following birth (Figs. S1, S2, Supplementary Data 1). Consistent with previous scientific literature focusing on infant community state types (ICSTs)[23], the species-level taxonomic classification of the sequenced reads revealed that *Bifidobacterium longum* and *Escherichia coli* were the most prevalent microbial infant gut components at the pre-weaning stage, followed by *Bifidobacterium pseudocatenulatum*, *Bifidobacterium breve*, *Collinsella aerofaciens*, and *Bifidobacterium bifidum* (Supplementary Data 2). Furthermore, as reported previously, the relative average abundances of these species decreased in post-weaning (Kruskal–Wallis test, *p*-value < 0.01), simultaneously with the progressive colonization by adult-associated bacterial species, including *Eubacterium rectale*, *Faecalibacterium prausnitzii*, *Ruthenibacterium lactatiformans*, *Akkermansia muciniphila*, and members of the *Bacteroides* genus, such as *Bacteroides uniformis* (Fig. S2, Supplementary Data 2)[23,24].

With the aim of investigating strain dynamics in the developing infant gut ecosystem, a total of 63 metagenomically assembled genomes (MAGs), corresponding to the 11 above-listed main gut-associated microbial species (Supplementary Data 3), were coupled with conspecific publicly available genome sequences in order to build 11 species-specific databases of reference strains (Supplementary Data 4). Following assessment for completeness (>90%) and ANI-driven dereplication, these 11 databases were employed to investigate strain-specific persistence and stability, i.e., the time span during which longitudinal samples harbored identical strains (Fig. S3)[25].

Collected data revealed that strains belonging to species associated with the introduction of solid diets, such as *E. rectale* and *C. aerofaciens*, assemble into heterogeneous strain communities, appearing vulnerable to intestinal niche changes that occur during the first two years of infant life (Fig. S3). Conversely, specific *B. longum*

subsp. *longum*, *B. bifidum*, *B. breve*, and *B. pseudocatenulatum* strains established stable host-microbe symbiotic relationships lasting beyond the weaning phase in 91%, 72 %, 54.5 %, and 45.4% of the inspected infants, respectively (Table S2, Fig. S3), thereby representing the most resilient, unvarying, and stable bacterial communities of the assessed infant gut microbiome (See supplementary text for details). Specifically, after accounting for sequencing depth, an adjusted average number of 1.58 *B. longum* subsp. *longum*, 1.44 *B. bifidum*, 0.82 *B. breve*, and 0.80 *B. pseudocatenulatum* strains were found to be shared among multiple time points within the first birth year (Fig. S3), thus emerging as significantly more persistent than other principal members of the suckling infant gut microbiome (Kruskal–Wallis with Dunn's post-hoc test, *p*-values after the Benjamini-Hochberg correction <0.05; Table S5; Fig. S4). Moreover, specific *B. bifidum* and *B. longum* subsp. *longum* strains were detected in the infant gut up to the second year after birth, coinciding with the last sampling time (Supplementary Data 5). Consistently, these bifidobacterial species are known to be genetically adapted to colonizing the infant's intestine due to particular metabolic activities and cooperative trophic interactions, i.e., cross-feeding actions, toward complex carbohydrates, such as HMOs and human mucin[26–28].

The strain-resolved dynamic of the decoded infant gut microbiome was experimentally validated using strain-specific primers through quantitative real-time PCR (qRT-PCR) (Supplementary Data 6), confirming that fluctuating and/or persistent patterns occurring during the infant gut microbiome development are strictly dependent on the microbial species.

To validate the results from our population-wide metagenomic study and to ensure that the bioinformatic approaches did not bias the observed bacterial persistence patterns, an independent validation cohort was constructed employing a large, publicly available infant dataset that includes fecal samples from multiple time points spanning the first year after birth[29] (Supplementary Data 1). The strain-resolved microbial community composition of this particular metagenomic dataset was determined employing the pipeline implemented by Mäklin et al.[30]. As detailed in Table S7, the development trajectories of the infant gut microbiome observed within the validation cohort confirmed what we noted in our study population, highlighting a longer-term persistence of bifidobacterial strains compared to those belonging to non-bifidobacterial species, i.e., *C. aerofaciens*, and *E. coli* (Chi-Squared post-hoc test, *p*-value < 0.05) (Supplementary Data 7). Moreover, a statistically significant higher persistence of the early *B. bifidum* and *B. longum* colonizing strains was observed in vaginally delivered infants when compared to those born by C-section (Chi-Squared test, *p*-values < 0.05) (Supplementary Data 7). These results expand on our previous findings, suggesting a potential greater ability of maternally derived bifidobacterial strains to persist in the infant gut microbiome during the neonatal period.

### Correlation between microbial resilience and host sex

As we mentioned above, the first seeding of (members of) the gut microbiota is believed to occur during delivery, involving the transfer of microbial lineages from the mother to the respective newborn[22,31]. In this context, we questioned if microbial strain persistence is more effective in females to sustain the intergenerational transmission of ecologically well-adapted gut microbiota members. For this purpose, the vaginally delivered fraction of the above-mentioned longitudinal metagenome infant dataset (72 females and 73 males, for a total of 145 vaginally delivered infants) were integrated with 357 additional, publicly available shotgun samples from longitudinal studies of the infant gut microbiome (54 females and 59 males), encompassing pre-weaning (0–6 months) and post-weaning (over six months) time-points (Fig. S1, Supplementary Data 1)[11,32]. All infants were healthy, delivered vaginally at term, and were not subjected to antibiotic treatment (Supplementary Data 1). Hence, considering the above-

described high persistence of bifidobacterial strains throughout the transition phase of weaning, we evaluated the sex-specific stability of strain communities belonging to *B. bifidum*, *B. longum* subsp. *longum*, *B. breve*, and *B. pseudocatenulatum* between pre- and post-weaning stages by using the StrainGE tool (see Methods).

Specifically, for each infant, the bifidobacterial strain dominating the gut microbiome at 0–6 months of age was compared with those preeminent after the introduction of complementary solid foods (around 12 months). At the species level, *B. bifidum* was detected from lactation to post-weaning in 31% of the 258 inspected infants, including 36 females and 44 males (Fig. 1a, Supplementary Data 9). While no sex-related difference in species-level stability was observed across weaning (Fisher test, *p*-value = 0.349), the dominant *B. bifidum* reference strains detected at 0–6 months persisted after the weaning phase in 24 out of 36 female infants (67%) and in 15 of the 44 males (34%) (Fig. 1a, Supplementary Data 9), suggesting greater strain-level stability in the large intestine of female infants (Fisher test, *p*-value = 0.007). Similarly, 49.6% of all assessed infants (67 females and 61 males) exhibited cross-weaning persistence of the *B. longum* subsp. *longum* species, whose dominant reference strains identified at 0–6 months were also found at 12–24 months in 45% of the females, being significantly higher than what was observed for male infants (21%) (Fisher test, *p*-value = 0.005) (Fig. 1a, Supplementary Data 9). Specifically, males appear to undergo higher fluctuations in (bifido)bacterial strain composition compared to female infants, whose early-engrafted dominant *B. bifidum* and *B. longum* subsp. *longum* strain was maintained through the weaning phase with a higher frequency.

In contrast, *B. breve* and *B. pseudocatenulatum* species were detectable across time-points (pre- and post-weaning) only in 14.3% and 9.6% of infants, respectively (Fig. 1a), showing no significant sex-associated difference in strain persistence (Fisher test *p*-value > 0.05) (Fig. 1a, Supplementary Data 9). As *B. longum* subsp. *longum* exhibited higher sex-biased stability/persistence in the infant gut, we extended the above-described analysis of dominant strains by exploring the whole *B. longum* subsp. *longum* strain composition throughout the weaning phase by using different bioinformatic approaches. For this purpose, we obtained 377 non-redundant *B. longum* subsp. *longum* genomes by assembling metagenome-derived data from the infant longitudinal datasets (258 infants, 126 females, and 132 males). This genome collection was integrated with publicly available *B. longum* subsp. *longum* chromosomal sequences and, following completeness assessments and dereplication, was employed as a genome database for the inStrain tool[33]. Considering the infants detected by inStrain with cross-weaning persistence of *B. longum* subsp. *longum* at the species level (71 females and 65 males), we observed that at least one strain found in the pre-weaning age was maintained in the post-weaning phase in 52 females (76%) and 31 males (47%) (Fisher test *p*-value = 0.003) (Fig. 1b), thus corroborating the findings reported above.

Moreover, after a persistence event (*n* = 83), the early-engrafted *B. longum* strains reached an average relative abundance of 71% ± 21% in the overall post-weaning *B. longum* subsp. *longum* strain communities (Fig. 1c). These figures, although similar between infant males and females, were significantly higher than those calculated for the *B. longum* subsp. *longum* strains not involved in persistence episodes (*n* = 171, average relative abundance of 29% ± 29%, Mann–Whitney test, *p*-value = 0.001) (Fig. 1c). These findings imply that when a persistence event occurs, it involves strains ecologically favored to colonize the infant gastrointestinal tract at higher relative abundance, thus dominating the conspecific strain population.

Notably, analysis of the overall *B. longum* subsp. *longum* strain population showed that persistence of specific strains seems to occur at a statistically significant higher rate in female infants compared to male infants, also highlighting a superior colonization capability of the persistent *B. longum* subsp. *longum* members compared to the coexisting non-persistent strains.

Based on current scientific data, one may argue that following initial inoculation by the maternal fecal bacteria, diet ingredients such as host-derived glycans play a role in selecting persistent (bifido)bacterial strains, such as *B. bifidum* and *B. longum* subsp. *longum* members, based on their ability to metabolize these amino sugars[34,35]. Such bifidobacterial strains first forage on lactose, HMOs, and perhaps other milk-associated glycans or glycoproteins, subsequently taking advantage of intestinal mucin glycans both as binding sites and as a carbon source in a strain-specific manner, leading to their long-lasting persistence[35].

## Identifying genes associated with sex-specific microbial persistence

As *B. bifidum* and *B. longum* subsp. *longum* strains exhibited sex-specific resilience across infant weaning, we performed a comparative genomic analysis to explore the potential genetic features distinguishing these bifidobacterial species from *B. breve* and *B. pseudocatenulatum*, which displayed no or poor persistent behavior or sex-related differences (Fig. S1). To gain an accurate overview, the analysis encompassed all complete and well-annotated genome assemblies available from public repositories for a total of 119 genomes belonging to these bifidobacterial taxa.

From these surveys, 14 protein families (referred to as PDC) were identified within the core/accessory gene repertoire unique to *B. bifidum* and *B. longum* subsp. *longum*, while they were absent in *B. breve* and *B. pseudocatenulatum* chromosomes (See Supplementary text for details) (Fig. 1b, Supplementary Data 10). Among these, we found two GH-encoding genes that were selected for further exploration considering the known importance of host glycans in modulating host-microbe interactions, i.e., the predicted extracellular membrane-anchored mucin-degrading glycosyl hydrolase family 101 (GH101)[34], and the glycosyl hydrolase family 136 (GH136) which is predicted to act as an extracellular lacto-N-biosidase[36] (Fig. S5). Specifically, genes encoding GH101 and GH136 enzymes were found in all screened *B. bifidum* genomes, while 89% and only 38% of the chromosomes belonging to *B. longum* subsp. *longum* showed the presence of genes specifying GH101 and GH136, respectively (Fig. 1b; Fig. S6; Supplementary Data 10). The involvement of GH101 and GH136 activities in host glycan metabolism was verified through a transcriptomic survey on *B. bifidum* PRL2010 cultivated under in vitro conditions, showing the up-regulation of these GHs when mucin was used as the sole carbon source rather than glucose (MRS-based medium) (Supplementary Data 11) (See supplementary text for details). These findings suggest that the GH101 and GH136 enzymes, acting on mucin glycan core structures, are involved in the observed long-term colonization in the host gut of *B. bifidum* and *B. longum* subsp. *longum* strains by providing an endogenous source of nutrients in the absence of dietary glycans.

## Sex-specific gut persistence of *B. longum* and *B. bifidum* strains from birth to later stages of host life

To validate the hypothesis that *B. bifidum* and *B. longum* subsp. *longum* strains can achieve higher colonization in the human (female) gut across human lifespan, a total of 12,415 cross-sectional metagenomic fecal samples (of which 6545 or 54% were of female origin and 5870 or 46% were derived from males) from roughly 3000 infants (0–4 years old, 1456 female and 1541 males), 918 children (5–18 years old, 434 females and 484 males), 6147 adults (19–55 years old, 3379 females and 2768 males), and 2353 elderly (56–90 years old, 1276 females and 1077 males) were subjected to strain-level analyses employing the same approach and involving the above described *B. longum* subsp. *longum* and *B. bifidum* reference genome databases in order to analyze the longitudinal infant data (Fig. S1). As expected, suckling infants showed the highest prevalence of *B. longum* subsp. *longum* and *B. bifidum* species, exceeding 70% and 40%, respectively (Fig. 2a, Supplementary

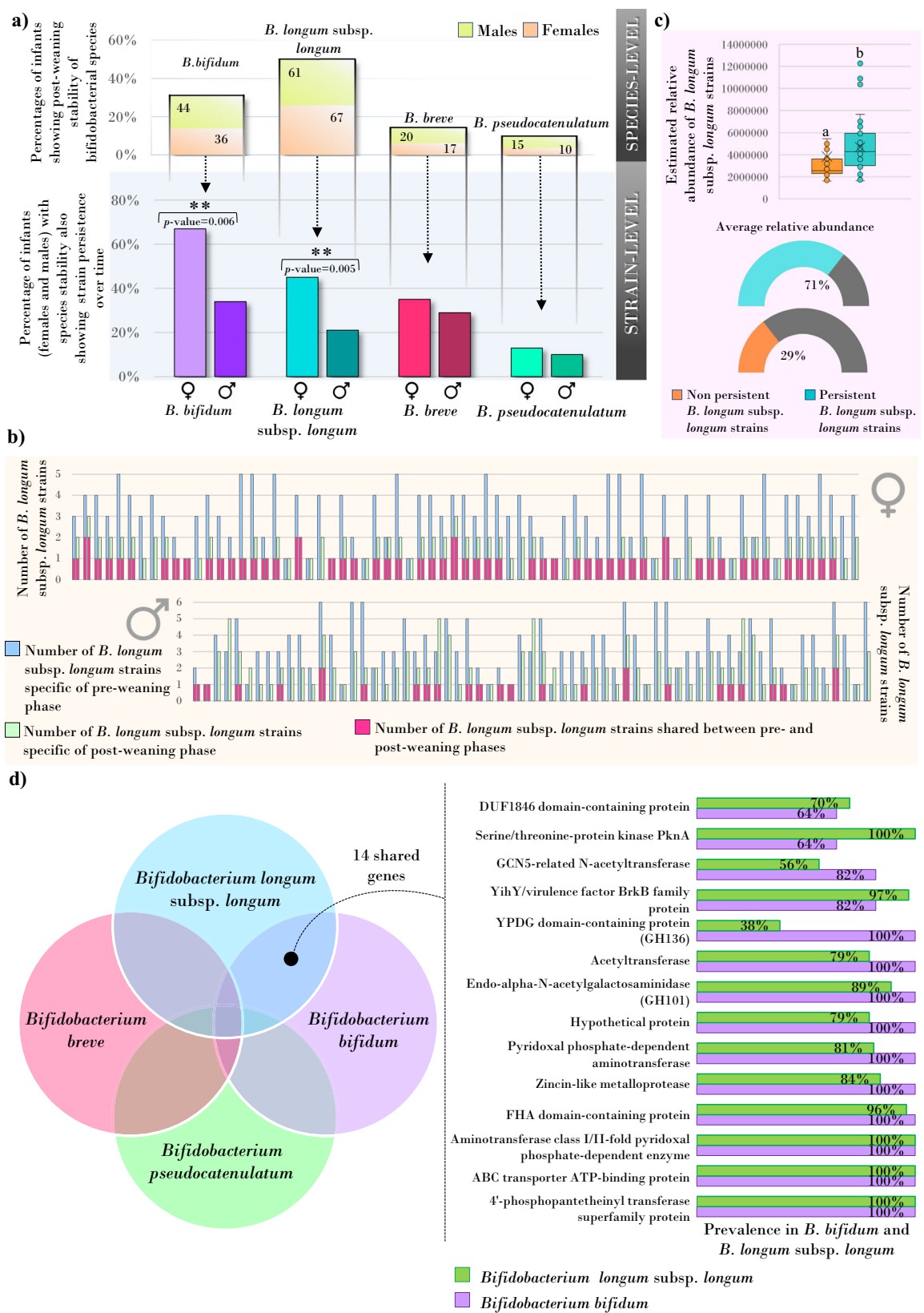

Data 10). Interestingly, when we explored adult populations, we observed that sex markedly impacted the prevalence of these bifidobacterial taxa (PERMANOVA $R^2 = 0.0627$ and $0.0558$; $F = 494.18$ and $420.03$, both $p$-values = $0.0099$; Supplementary Data 12). Specifically, 41% and 28% of the adult female subjects harbored members of *B. longum* subsp. *longum* and *B. bifidum* species, respectively, which,

conversely, colonized only 26% and 11% of the age-matched male individuals (Fisher test, $p$-values < 0.001) (Fig. 2a, Supplementary Data 10). In contrast, elderly subjects showed the lowest prevalence of bifidobacterial species, ranging from an average of 26% for *B. longum* subsp. *longum* to an average of 9% for *B. bifidum* species, with similar values between female and male individuals (Fig. 2a, Supplementary

**Fig. 1 | Gender-specific persistence of *B. bifidum*, *B. longum* subsp. *longum*, *B. breve*, and *B. pseudocatenulatum* species and the 14 genes shared between the female-associated persistent bifidobacterial species.** In panel (**a**), the bar chart on the top displays the species-level persistence from pre- (1–6 months) to post-weaning stages (12–24 months) of *B. bifidum*, *B. longum* subsp. *longum*, *B. breve*, and *B. pseudocatenulatum* in the infant population. Below, bar plots represent the 12–24-months gender-specific stability of the bifidobacterial strains, expressed as the percentage of (infant) females and males showing after-weaning persistence of the same bifidobacterial strains identified at 0–6 months. Statistically significant gender-related differences were highlighted with an asterisk on the top of the columns (Fisher test, *p*-value = 0.006 and *p*-value = 0.005). Panel (**b**) refers to the inspection of the whole *B. longum* subsp. *longum* strain communities in female and male infants. Each pair of bar plots shows the number of strains identified in the pre-weaning (left) and post-weaning phase (right). Different colors highlight the number of *B. longum* subsp. *longum* strains found only in pre-weaning age (light blue), only in the post-weaning phase (light green), and shared between pre- and post-weaning time-point (pink). Panel (**c**) depicts the statistically significant difference in an estimated relative abundance (obtained by normalizing the genome coverage on the corresponding genome length) of persistent (*n* = 83, light blue) and non-persistent (*n* = 171, orange) *B. longum* subsp. *longum* strains in the infant post-weaning time points (Mann–Whitney test, *p*-value = 0.001). The boxes are determined by the 25th and 75th percentiles. The whiskers are determined by 1.5 interquartile range (IQR). The line in the boxes represents the median, while the cross marker (X) represents the average. In panel (**d**), the Venn diagram on the left highlights the 14 genes shared by *B. longum* subsp. *longum* and *B. bifidum*, while bar charts on the right-side report the prevalence of each PDC in publicly available complete genomes *B. longum* and *B. bifidum*.

Data 10). Possible confounding factors such as dairy food consumption and lactase persistence were accounted for the differences in *B. longum* subsp. *longum* and *B. bifidum* prevalence between adult females and males. Notably, since this type of information was not available in public datasets, we used geographic regions as proxy variables (south/north Europe for lactase persistence and Europe/China for dairy food consumption), evidencing no significant association between geographic region and sex-dependent prevalence of the target (bifido)bacterial species (Fisher test *p*-values > 0.05) (Supplementary Supplementary Data 13).

These findings support the intriguing notion that *B. bifidum* and *B. longum* subsp. *longum* can stably colonize the human gut from infancy to adulthood with an apparent preference in women during their reproductive age, possibly as potential reservoirs for microbial transmission to new generations.

The genomes of *B. bifidum* and *B. longum* subsp. *longum* with completeness equal to or greater than 90% detected within the infant, child, adult, and elderly gut microbiomes were subjected to a genome-wide screening to assess the occurrence of genes encoding predicted GH101 and GH136 enzymes and thus to deduct their prevalence across populations (Fig. 2b, Supplementary Data 10). According to the survey results, the GH101 of *B. bifidum* and *B. longum* subsp. *longum* were detected between 88% and 100% of the 12,415 fecal metagenomic datasets, which, being greater than expected from public genome screening, supports their potential key role in colonization and survival of the human gastrointestinal tract across the entire host lifespan (Supplementary Data 10).

Instead, the GH136 was found in an average of 45% of metagenomic samples, with the highest frequency found in adult women (63%), which is 50 % higher than that observed in age-matched males (42%) (Fisher test, *p*-value < 0.05) (Supplementary Data 10). However, such sex-specific differences were not evident in early infancy and older adulthood. Intriguingly, the higher colonization performances observed in females compared with males seem to disappear in individuals older than 50 years, which is concordant with menopause age (Supplementary Data 10). Recently, sex hormones have been regarded as potent drivers of sexual dimorphism at the gut microbiome level, being associated with compositional differences between sexes and profound changes in the gut microbial community of pregnant women[37–40]. It has been proposed that, besides its role in the modulation of the immune system and bile acid secretion, which may then regulate the gut microbiota, steroid sex hormones can be metabolized by specific gut-associated bacterial enzymes, thus directly impacting microbial metabolism and growth[37,41,42]. Although little is yet known about the impact of sex on the human gut microbiota and even less about the underlying mechanisms, it has been reported that a higher level of sialylation characterizes the intestinal mucus of the female gut compared to that of the male population[43]. Furthermore, it has also been argued that the female sex hormone estradiol may upregulate the expression and glycosylation of human

mucins[44–46], possibly shaping the female gut microbiota in favor of mucin-utilizing (bifido)bacteria. Interestingly, the estradiol level is higher in females than males, even in the prepubertal phase[47]. Indeed, the hypothalamic-pituitary-gonadal axis, which is involved in the development and regulation of the reproductive system, undergoes transient activation during the first six months of life in males and the first two years in females[48,49]. This event, called minipuberty, or "endocrine puberty", induces testosterone production in males and estradiol in females[50–52].

In addition to these hormone-driven effects, it should be noted that complex mechanisms of DNA methylation patterns in colon tissue-specific genes (excluding loci located in the X and Y chromosomes) may contribute to the establishment of sex- and age-related differences in the intestinal environments[53,54]. Consistently, variability in methylation signatures was observed in subjects of various ages, including both newborns and adults, when comparing females to males[53,55].

Altogether, these findings suggest that, compared to males, the female-specific intestinal surroundings, including glycan structures, may be structurally more predisposed to creating a suitable environment for certain early colonizing microbial species, such as *B. bifidum* and *B. longum*. In particular, when the gene encoding the GH136 enzyme is present, it could play a part in enhancing (bifido)bacterial persistence in women of reproductive age, possibly due to a sex-specific mucus structure.

## The role of GH136 in bacterial vertical transmission from mother to newborn

*B. bifidum* and *B. longum* represent two of the main species that can be maternally inherited by means of vertical transmission[56]. In particular, colonization of the infant gut from maternal gut-associated strictly anaerobic species, such as *Bifidobacterium*, is believed to occur through direct contact with maternal gut microbiota during birth and/or involve an entero-mammary pathway, which may transfer ecologically well-adapted bacteria via breastmilk to infants[57–60]. Starting from this scientific evidence and exploiting the non-ubiquitous presence of GH136-encoding genes in *B. longum* strains, we decided to estimate the involvement of GH136 in mother-to-infant vertical transmission events through inspection of publicly available metagenomic fecal samples from 132 full-term vaginally delivered healthy newborns and their corresponding mothers[5,61] (Fig. S1). Notably, fecal samples collected from mothers at childbirth and newborns at one month were subjected to *B. bifidum* and *B. longum* subsp. *longum* strain tracking analyses, and the reads were then mapped to the GH136 gene sequence.

Our results indicated that specific strains of *B. longum* subsp. *longum* and *B. bifidum* were present in the samples from mother-infant dyads in 36.4% and 29.5% of the screened cases, respectively, suggesting perinatal vertical transmission events. Interestingly, 72.9% of the *B. longum* subsp. *longum* strains detected as vertically transmitted from mothers to newborns harbor the GH136-encoding gene, while

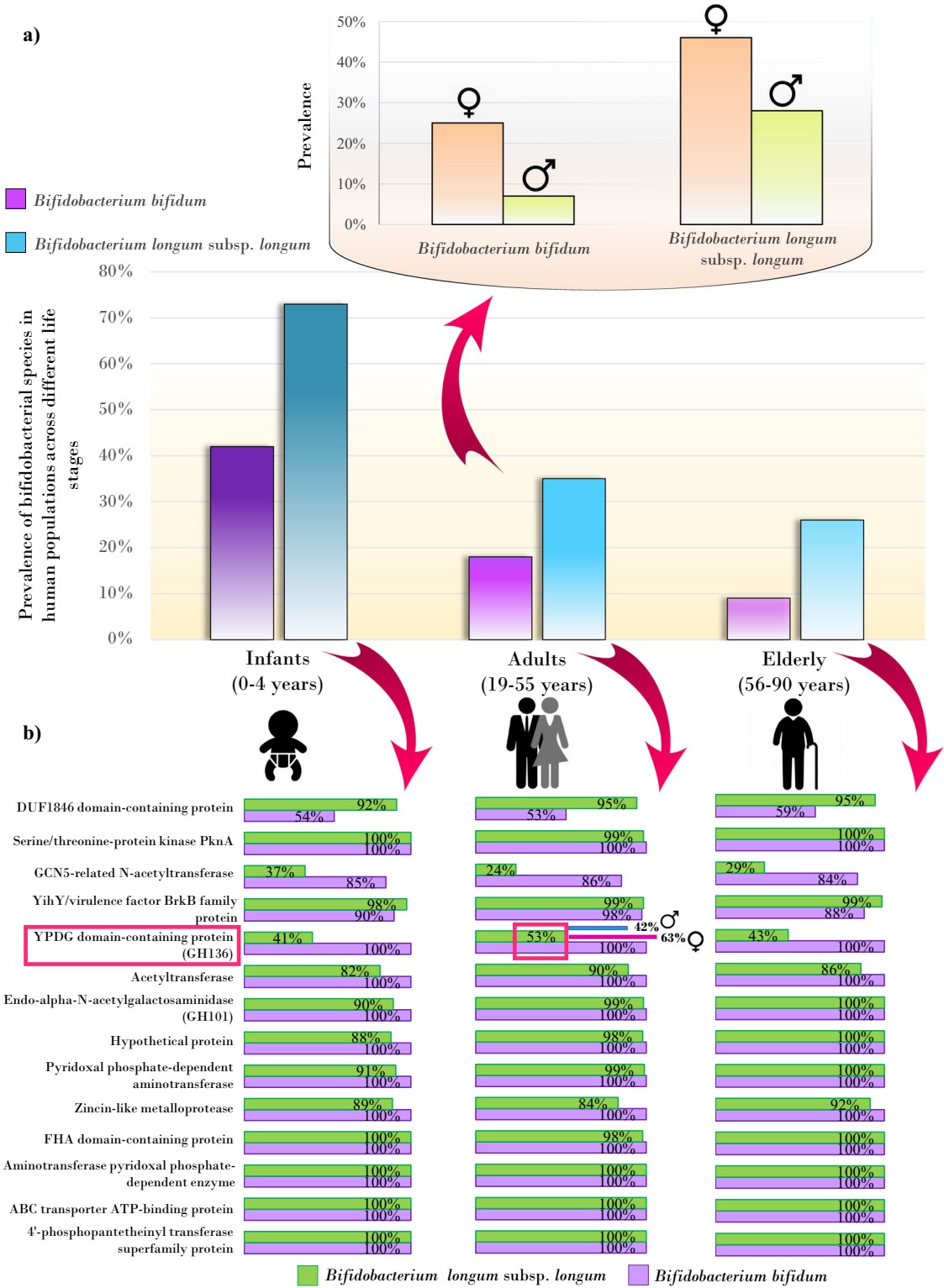

**Fig. 2 | Persistence of *B. bifidum* and *B. longum* subsp. *longum* across human life stages.** In panel (**a**), the vertical bar graph depicts the persistence of *B. bifidum* and *B. longum* subsp. *longum* across the human life span. On the top, bar plots show the differences in the persistence of these bifidobacterial species between female and male individuals aged 5–55 years. In panel (**b**), horizontal bar charts represent the prevalence of the 14 genes shared by *B. bifidum* and *B. longum* subsp. *longum* across populations. Statistically significant gender-based differences in the occurrence of GH136 are highlighted with a red frame, and the respective prevalence (percentage) are reported alongside (Fisher test, *p*-value < 0.05).

just 28.6% of the *B. longum* subsp. *longum* strains that did not appear to be involved in vertical transmission events were shown to possess the GH136 gene (Fischer Test *p*-value < 0.001, Supplementary Data 14). Accordingly, sex-related genetic and epigenetic host factors seem to be associated with specific microbial genomic features to ensure persistence and, therefore, the establishment of selected consortia of bifidobacterial strains that may be maternally transmitted through delivery to the offspring.

## Retrospective clinical studies support the GH136-driven, long-term gut persistence of *B. longum* subsp. *longum* in women

In order to assess the role of the accessory GH136 in the colonization and persistence of *B. longum* subsp. *longum* in the human gut, we analyzed data from published retrospective clinical studies in which healthy human participants received daily oral doses of viable cells of two genetically different *B. longum* subsp. *longum* strains[62,63] (Fig. S1). Specifically, a total of 21 healthy individuals (52% females) received a treatment consisting of a daily dose of $10^{10}$ viable cells of *B. longum* subsp. *longum* AH1206, a strain that harbors genes encoding the GH136 and GH101 enzymes (Supplementary Data 10). A second group of 10 individuals (40% females) consumed the same daily dosage ($10^{10}$ viable cells) of a *B. longum* subsp. *longum* strain named AG1, possessing GH101 but lacking the GH136 gene (Supplementary Data 15).

Shotgun metagenomic data of stool samples collected before the intervention (baseline), after 21–28 days of oral bacterial administration (treatment), and after the follow-up period (persistence) were exploited to evaluate the persistence of *B. longum* subsp. *longum* AH1206 and AG1 strains (Fig. 3a).

Fecal metagenomic reads from each subject were mapped against the administered bifidobacterial strain genome sequences (AH1206 or AG1), considering only >99% homology matches. The results showed that both AH1206 and AG1 strains were detectable at the end of the 28-day treatment period and then decreased after the termination of microbial supplementation in a strain-specific manner (Fig. 3b, Supplementary Data 15). Remarkably, the relative abundance of AH1206 mapped reads remained significantly higher compared with the pre-treatment baseline even 200 days after completion of treatment (Wilcoxon test, *p*-value < 0.01) (Fig. 3b, Supplementary Data 15). In contrast, the relative average abundance of AG1 mapped reads was not significantly higher when compared to the situation before the microbial intervention as early as 28 days after interruption of consumption (Wilcoxon test, *p*-value > 0.05), indicating that the level of AG1 one month after treatment was reverted to that observed in the pre-treated microbiomes. Intriguingly, when assessing the degree of long-term AH1206 colonization among female and male volunteers, we identified gender-related differences in the level of strain persistence. Indeed, at the persistence test time-point (around 200 days of follow-up), the average relative abundance of AH1206-related mapped reads from females was found to be significantly higher compared with their own baseline (Wilcoxon test, *p*-value < 0.01), while male participants did not exhibit such long-term persistence of AH1206 (Wilcoxon test of average mapped reads at 200 days vs. baseline, *p*-value > 0.05) (Fig. 3c, Supplementary Data 15).

Overall, these findings possibly indicate that GH136 positively impacts the (female host-associated) persistence of *B. longum* subsp. *longum* strains.

## Evaluation of the molecular interaction of persistent and non-persistent *B. longum* strains with human intestinal cells through transcriptomics analyses

To assess the role of the non-ubiquitously present *B. longum*-encoded GH136 in host-microbe cross-talk and to investigate molecular interactions between persistent and non-persistent *B. longum* subsp. *longum* strains and the host cells, we applied an in vitro approach involving human cell lines placed in contact with bacterial cells (Fig. S1). Specifically, we cultivated Caco2/HT29-MTX cell monolayers in direct physical contact with *B. longum* subsp. *longum* PRL2022 or 1898B strains, which had been selected based on the presence or absence of the gene encoding the accessory GH136, respectively. Subsequently, the transcriptomes from both bacterial and human cell lines were investigated through RNA-Seq experiments aimed at evaluating the differentially expressed genes (DEGs) between each treatment (PRL2022- and 1898B-Caco2/HT29-MTX contact) and the respective control conditions (absence of contact), considering

statistically significant a fold-change ≥ 2 at a *p*-value ≤ 0.05 after correction for multiple comparisons using the False Discovery Rate (FDR) procedure (Figs. S7, S8, Table S16, Supplementary Data 17).

Following host cell contact, a total of 334 and 492 bacterial genes from PRL2022 and 1898B strains, respectively, were classified as DEGs when compared to control samples (Supplementary Data 16). Among the statistically significant upregulated transcriptomes (~50% of total DEGs) (Fig. 4a, Fig. S6), we found transcripts corresponding to genes encoding priming and processing glycosyltransferases involved in exopolysaccharide (EPS) production, several carbohydrate and amino acid modifying enzymes, and various protein kinases (Supplementary Data 16). Focusing on (bifido)bacterial host-glycan and carbohydrate metabolism genes, it was found that both PRL2022- and 1898B-related transcriptomes showed significantly higher expression of the GH101-corresponding gene (JL750_RS06690, BLSL_RS08555) when compared to control samples, in addition to genes predicted to encompass a number of 16–19 ABC transporters involved in carbohydrate uptake, including that identified above among the 14 genes shared between *B. longum* and *B. bifidum* (Fig. 4a, b, d, e; Fig. S7). Moreover, the GH136 gene (JL750_RS09800), present only in *B. longum* subsp. *longum* PRL2022, was shown to be upregulated (Fig. 4b, Supplementary Data 16).

As a result, intestinal bacterial colonization appears to be a multifactorial process that involves various carbohydrate-modifying enzymes and microbial surface components. These appear to corroborate the involvement of the (bifido)bacterial mucin-degrading enzymes GH101 and GH136 in host-microbe interplay and mucosal surface colonization. However, as mammalian hormones may affect bacterial gene expression[64–66], we evaluated whether the endocrine milieu contained in the fetal bovine serum (FBS, used in combination with DMEM for human cell culture, see the "Methods" section) could impact the expression of the GH136 gene. Specifically, we differentially grew PRL2022 on the culture medium DMEM with and without FBS, obtaining no significant changes in RT-qPCR-based GH136 gene expression results (*t*-test *p*-value > 0.05).

Furthermore, the transcriptome of human cell lines placed in contact with *B. longum* subsp. *longum* was assessed and compared with that achieved in the absence of bacterial cells. Notably, a total of 1253 (874 upregulated) and 1404 (892 upregulated) host cell-related genes were identified as DEGs in the PRL2022-exposed and 1898B-exposed groups, respectively (Fig. S8, Supplementary Data 17). Among the *B. longum*-induced upregulated host transcripts, we found genes encoding pattern recognition receptors, which react to bacteria (e.g., Toll-like receptors)[67,68], and cell signaling molecules that aid cell communication in immune responses (e.g., cytokines)[69] (Supplementary Data 17).

Focusing on the expression of human mucin genes, transcriptome analysis of Caco2/HT29-MTX cells upon exposure to *B. longum* subsp. *longum* PRL2022 (which harbors a GH136-encoding gene) revealed upregulation of genes encoding mucin5B (mucus/gel-forming), mucin3A, and mucin17 (cell surface-associated)[70], as well as glucosaminyl (N-acetyl) transferase 3, which catalyzes the formation of core 2 and core 4 O-glycans on mucin-type glycoproteins[71,72] (Fig. 4c, Supplementary Data 17). Remarkably, only mucin3A and mucin5B were expressed at significantly higher levels in the non-persistent *B. longum* subsp. *longum* 1898B-exposed host-derived transcriptome compared with control samples (Fig. 4f, Supplementary Data 17).

Overall, these findings suggest that *B. longum* subsp. *longum* strains significantly influence the transcriptome of human intestinal cells, modulating the expression of genes involved in the synthesis of mucus layer components. Remarkably, *B. longum* subsp. *longum* PRL2022 (encoding the GH136 enzyme) appeared able to stimulate the expression of host mucin and mucin-related genes to a greater extent (100% increase in the number of upregulated host mucin-related genes) than the conspecific 1898B strain lacking the GH136 gene, implying a possible competitive advantage and enhanced persistence in the gut of (female) human hosts.

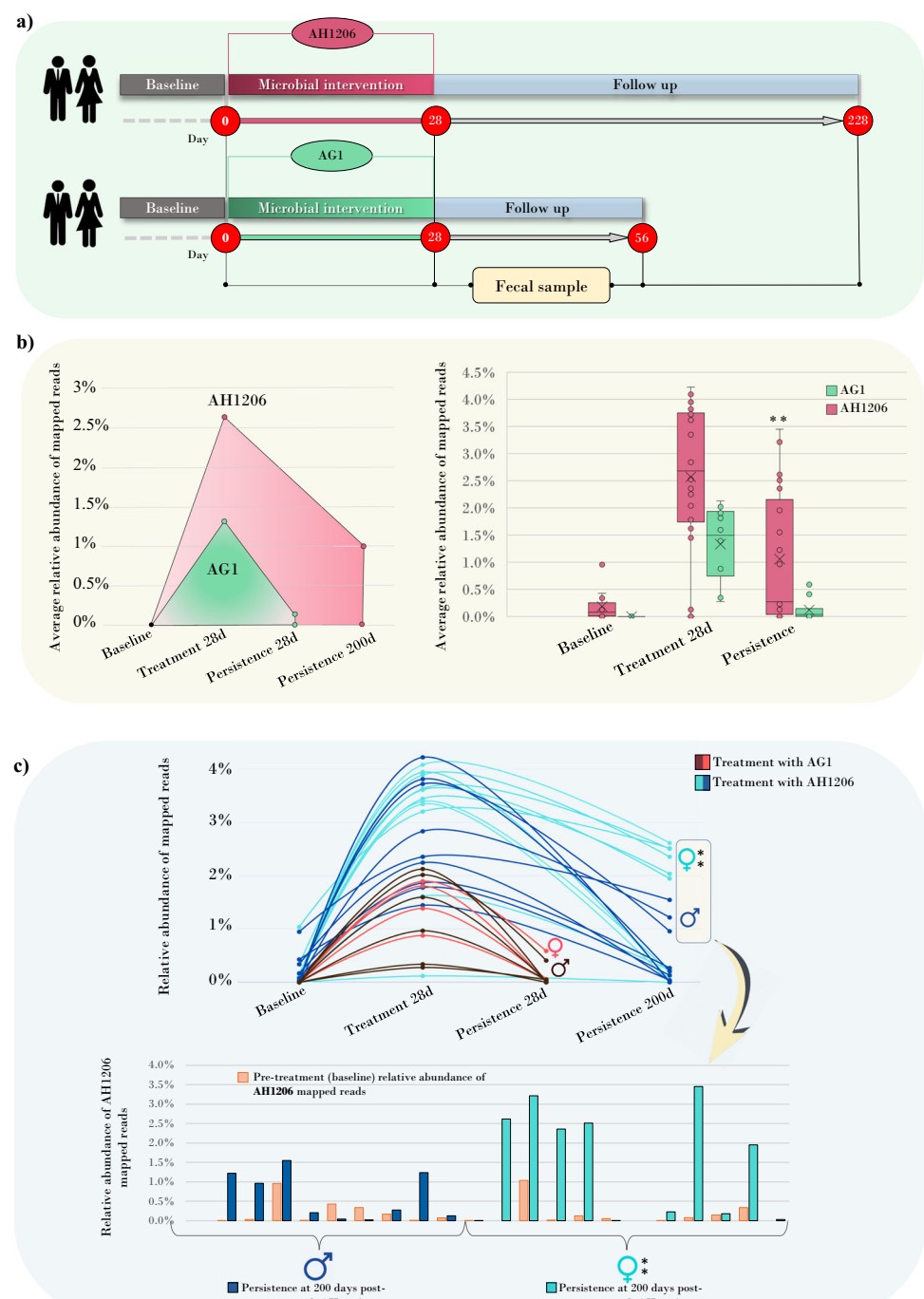

**Fig. 3 | Analysis of data from human retrospective clinical studies based on the supplementation of *B. longum* subsp. *longum* AH1206 and AG1 strains.** Panel (**a**) shows the experimental outline of the human retrospective trials considered in this study. Panel (**b**) reports the average relative abundance of mapped reads of *B. longum* subsp. *longum* AG1206 and AG1 in the metagenomic samples during the bacterial supplementation and follow-up. The significant difference in the average abundance of AH1206 and AG1 mapped reads between follow-up (persistence) and the corresponding baseline is indicated by an asterisk in the Box and Whisker plot (**\*\****p*-value < 0.01; Wilcoxon signed-rank test, *p*-value = 0.009). The boxes are determined by the 25th and 75th percentiles. The whiskers are determined by 1.5 interquartile range (IQR). The line in the boxes represents the median, while the cross marker (X) represents the average. In panel (**c**), the trend of AG1206 and AG1 mapped reads (relative abundance) across the corresponding interventional studies (*n* = 22 and *n* = 10, respectively) is shown for each sample. Differences in the persistence of AH1206 strains between female and male populations (*n* = 21, 11 females and 10 males) are detailed in the bar plots depicting the average relative abundance of AH1206 mapped reads detected at baseline (*n* = 21, 11 females and 10 males) and after the termination of treatment (*n* = 21, 11 females and 10 males) (**\*\****p*-value < 0.01; Wilcoxon signed-rank test, *p*-value = 0.039).

## Discussion

Several studies have demonstrated that vertical transmission from mother to infants is a pivotal route for the early establishment of (certain) members of the gut microbiota, which can persist across subsequent stages of human life, although with a reduced abundance[8,28].

In this context, it is particularly important for the human female host to sustain the persistence of those early colonizers that may later be maternally transmitted to new generations.

In the current study, through multi-omics approaches, we revealed that *B. bifidum* and specific strains of *B. longum* subsp. *longum*

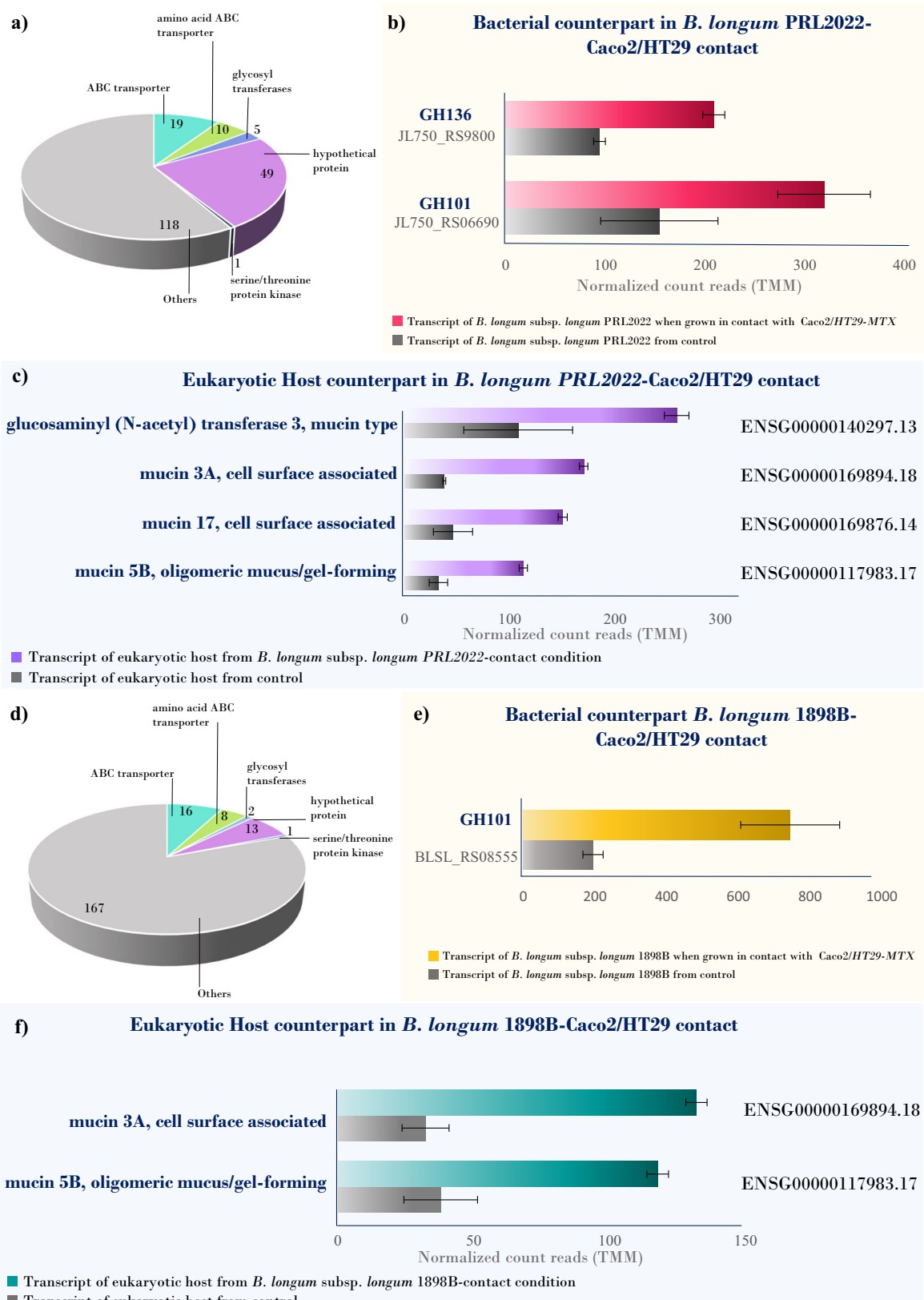

can establish a long-lasting colonization behavior preferentially in the intestinal environment of females compared with male individuals. Notably, these taxa are renowned for their highest level of vertical transmission among bifidobacterial species[22,73]. Interestingly, screening for the genetic determinants possibly involved in (female) human gut persistence led to the identification of two mucin-degrading GH families, GH101 and GH136, present in all *B. bifidum* and particular *B. longum* subsp. *longum* strains.

Specifically, the non-ubiquitous distribution of the GH136 gene within *B. longum* subsp. *longum* taxon highlights the involvement of this gene in mother-to-infant vertical transmission and led to its identification as a strain-specific genetic key determinant for stable

**Fig. 4 | Focus on differentially expressed bacterial mucin-degrading genes and host mucin-producing.** In panel (**a**), the chart graph highlights the number of different gene functional categories among the upregulated genes of *B. longum* subsp. *longum* PRL2022 when grown in contact with Caco2/HT29-MTX host cells. In panel (**b**), the horizontal bar plot shows the differentially expressed GH101 and GH136 mucin-degrading genes between *B. longum* subsp. *longum* PRL2022 grown in contact with Caco2/HT29-MTX human cells and control. In panel (**c**), the horizontal bar plot reports the mucin-producing genes found upregulated in the eukaryotic host transcriptome from *B. longum* PRL2022-exposed Caco2/HT29-MTX vs. control. Panel (**d**) depicts the number of different gene functional categories among the upregulated genes of *B. longum* subsp. *longum* 1898B when

grown in contact with Caco2/HT29-MTX host cells. Panel (**e**) shows the differentially expressed GH101 mucin-degrading genes between *B. longum* 1898B grown in contact with Caco2/HT29-MTX human cells and control. In panel (**f**), the horizontal bar plot reports the mucin-producing genes found upregulated in the eukaryotic host transcriptome from B. longum 1898B-exposed Caco2/HT29-MTX vs. control. Error bars represent standard deviations from three independent replicates. After normalization of row counts, genewise exact tests were computed to assess the differential expression of each gene. Adjustment of *p*-values for multiple hypotheses was performed through the false discovery rate (FDR) procedure.

female gut colonization. Recent studies have considered the importance of sex hormones in sex-dependent trajectories of the gut microbiome, besides the well-known environmental factors, age, dietary habits, geographical origin, and antibiotics[37–40,42]. Consistently, a higher level of sialylation, likely driven by sex-related hormones and potentially complex epigenetic mechanisms, was previously noted in the intestinal mucus of the female gut compared with that of the male population[43–46], suggesting that a sex-specific intestinal environment could be in a prime location to actively select persistent mucin-degrading (bifido)bacterial colonizers.

Overall, our findings propose intriguing and novel strict cooperation between host physiology and microbial genetics as a result of ancient (bifido)bacteria-human coevolution aimed at ensuring the maternal persistence of those microbial species that may at some point be vertically transferred to the next generation.

## Methods
### Study population
The study included 11 vaginally delivered infants born after an uncomplicated pregnancy and recruited at the Central University Hospital of Asturias (Northern Spain). The study was approved by the Regional Ethical Committee of Asturias Public Health Service (Ref.N° 51/18) and the Ethical Committee of CSIC (Ref 136/2018). Informed written consent was obtained from each infant's parent. Fecal samples were collected at scheduled appointments from one month to two years after birth (Supplementary Data 1). After sequencing (see below), the number of persistent strains detected in each post-weaning sample (corresponding to 12 or 24 months after birth) was normalized to account for differences in sequencing depth. Specifically, a normalizing factor was calculated for each sample as the ratio between the mean sequenced reads across the whole dataset and the number of sequenced reads in the sample. The obtained normalizing factor was then applied to the number of persistent strains identified in the metagenomic sample, thus obtaining an adjusted number of persistence strains.

A validation cohort was constructed to corroborate the findings observed in the study cohort employing a large and deep-sequenced publicly available longitudinal infant dataset[29] (PRJEB32631) (Supplementary Data 1). Specifically, infant fecal samples collected in the first 21 days after birth (pre-weaning phase) and after six months postnatally were selected as suitable for validating the (bifido)bacterial persistence pattern (Supplementary Data 1). This validation dataset was parsed by combining the recent mGEMS and mSWEEP methods[30,74–76], which assign short metagenomic reads to genomic bins corresponding to individual genomes of predefined species and estimate the relative abundances of reference bacterial strains, respectively. The k-mer-based pseudo-alignments against pre-build reference sequence databases were obtained through the Themisto software (version 3.1.2). Notably, this procedure was also employed to recover a set of reference genomes belonging to the target species characterizing the pre-weaning infant gut microbiome. In detail, a total of 371 MAGs belonging to *B. bifidum*, *B. longum* subsp. *longum*, *B. breve*, *B. pseudocatenulatum*, and *E. coli* were successfully reconstructed with

an average completeness degree of 86.7% ± 10% (based on checkM method) and were employed as reference genomes to confirm the within-host strain variation patterns observed throughout the weaning phase in the infant study population.

### Publicly available datasets
To extensively investigate the host sex-related bacterial persistence pattern, the newly sequenced longitudinal dataset was supplemented by 357 publicly available shotgun metagenomic samples from two studies that sampled 113 infants longitudinally[11,32]. In particular, we selected datasets corresponding to the analysis of the healthy infant gut microbiome at pre- (<6 months old) and post-weaning (>6–12 months old) development stages. In addition,, to create a comprehensive population-based cohort covering different age groups, cross-sectional fecal metagenomic samples of 12,415 healthy individuals aged from a few days after birth to 90 years were retrieved from 146 different publicly accessible studies. Specifically, only healthy (control) individuals and a single sample per each individual were retained from every study. A complete list of samples and metadata is available in the supplementary material (Supplementary Data 1). Notably, while this collected population dataset does not include subjects for whom any parental relationships emerged from corresponding metadata, shotgun data from mother-infant dyads were considered separately. Batch effects caused by different sequencing technologies were controlled by selecting only raw data produced through Illumina DNA sequencing platforms. Moreover, to overcome the potential confounding effects of library size, we randomly selected 5,000,000 reads from each sample, such that all samples have the same library size, while samples with fewer reads than 5,000,000 were discarded. Before applying this procedure, we used 250 deep-sequenced metagenomic samples (PRJEB32631) to compare the results obtainable by strain-level profiling using all the available reads (average of 8,826,752 ± 1,931,731 after *Homo sapiens* filtering) and a random subsample of 5,000,000 reads to ensure that no valid information on the microbial community structure was lost. The Mann–Whitney *U* test calculated on the strain profiles of four target species revealed comparable ability in detecting within-species variation in the metagenomic reads (Supplementary Data 8).

Confounding variables related to population structure, such as ethnicity/geographical origin and age, were tested through the permutation analysis of variance (PERMANOVA). Specifically, we stratified by age groups and assessed the statistical significance (*p*-value), the proportion of explained variance ($R^2$), and the effect size (*F* value) for each categorical variable. In addition, confounding effects of possible dairy food consumption and lactase persistence were tested by assuming geographical region as a proxy parameter. Specifically, subsets of metagenomic samples from Southern Europe (*n* = 417, 217 males and 200 females) and Northern Europe (*n* = 413, 186 males and 227 females) were selected to test the association between sex-dependent prevalence of *B. longum* subsp. *longum* and *B. bifidum* and lactase persistence, while fecal samples of subjects from China (*n* = 831, 413 males and 418 females) were compared with those from Europe (*n* = 830, 403 males and 427 females) to account for dairy consumption.

## Bacterial DNA extraction

Stool samples were stored on ice immediately after collection and shipped to the laboratory under frozen conditions, where they were preserved at −20 °C until processing. DNA extraction was performed using the QIAmp DNA Stool mini-kit according to the manufacturer's instructions (Qiagen, Germany). DNA quantification was achieved using the Qubit fluorometer (Thermo Fisher Scientific).

## Whole-genome sequencing and taxonomic classification

Shotgun metagenomic sequencing was performed by GenProbio srl (www.genprobio.com). DNA library preparation was performed using the Nextera XT DNA sample preparation kit (Illumina, San Diego, CA) according to the manufacturer's instructions. One ng input DNA from each sample was used for library preparation. The isolated DNA underwent fragmentation, adapter ligation, and purification. The ready-to-go libraries were pooled equimolarly, denatured, and diluted to a sequencing concentration of 2 pM. Sequencing was performed on a NextSeq 500 instrument (Illumina, San Diego, CA), according to the manufacturer's instructions, using the 2 × 150 bp High Output sequencing kit and spike-in of 1% PhiX control library. Whole-metagenome shotgun (WMGS) sequencing of the above-mentioned 43 infant gut microbiomes produced an average of 7,940,484 ± 2,078,529.565 paired-end 150 bp reads per sample. Following quality filtering (minimum mean quality score, 20; window size, 5 bp; and minimum length, 80 bp) and removal of reads that map on the *Homo sapiens* genome, an average of 6,355,063 ± 1,425,089 microbial reads per sample were retained (Supplementary Data 1).

Taxonomic profiling of sequenced reads, including those retrieved from publicly available shotgun datasets, was achieved with the METAnnotatorX2 bioinformatics platform[77], using the up-to-date RefSeq (genome) sequence database retrieved from the National Center for Biotechnology Information (NCBI) (https://www.ncbi.nlm.nih.gov/refseq/). Species-level taxonomic classification of each read was achieved through Megablast[78] (with option -e-value 1e-5, -qcov_hsp_perc 50) using >94% alignment identity. Reads that showed the same sequence identity against more than one bacterial species were discarded. Similarities between samples (beta-diversity) were computed by Bray-Curtis dissimilarity based on species abundance. PCoA representation of beta-diversity was performed using ORIGIN version 9.8.0.200 (https://www.originlab.com/2021).

## Bacterial genome assembly

For each unique host infant, representative genomes of 11 main gut-associated microbial species (Supplementary Data 3) were reconstructed as previously reported[17,77], leading to de novo metagenomics assembly and taxonomic classification of 67 bacterial genomes with a minimum average read coverage of 12X and a total genome size compatible with what was reported in literature. In detail, raw data of shotgun metagenomic sequencing (fastq files) that passed quality filtering and human genome mapping were used as input for SPAdes assembler v3.12[79], using default parameters enabling the metagenomic flag option (-meta). SPAdes parameters were combined with minimum k-mer sizes of 21, 33, and 55 to a maximum of 77, 99, and 127 based on the paired-end read length, as previously described[80]. Following assembly, ORFs of each assembled contig were predicted with Prodigal[81] with default parameters and then annotated using MEGAnnotator software[82,83].

## Construction of reference microbial genome databases and metagenomic strain-level analyses

To untangle strain communities and investigate strain dynamics in the infant, adult, and elderly gut microbiomes, 11 species-specific databases of reference genomes were constructed using 63 MAGs coupled with the genome sequences publicly available on the NCBI RefSeq database (complete and draft high-quality genome sequences with

<90 contigs) (Supplementary Data 4). Specifically, for each species, collected genomes (MAGs and publicly available) were processed through the open-source software Strain Genome Explorer (StrainGE) (github.com/broadinstitute/strainge) and dereplicated (clustered) according to the Average Nucleotide Identity (ANI) values (threshold of 99 % ANI value) (Supplementary Data 4). Subsequently fastq from shotgun metagenomic sequencing were analyzed using the Strain Genome Search Tool (StrainGST, an integrated component of the StrainGE tool suite)[84] that, following k-merizing both input fastq and reference genomes (straingst kmerize -k 23), iteratively compares the sample-associated k-mers set with those obtained from the reference genomes. As a result, StrainGE returns the reference genomes more similar (strongest StrainGST score, default threshold of 0.02, which is optimized to maximize sensitivity and minimize false positives) to those present within the metagenomic samples. This approach was used to investigate longitudinal strain stability in infants by comparing the strains profiled at 0–6 months (pre-weaning) with those identified at 12–24 months (post-weaning). To compare the whole *B. longum* subsp. *longum* strain communities between infant females and males throughout the weaning phase, we metagenomically assembled an additional 542 *B. longum* subsp. *longum* genomes from the 113 publicly available infant metagenomes (357 longitudinal samples) as described above. This collection was enriched with 404 *B. longum* subsp. *longum* genomes (complete and draft genomes with <90 contigs) retrieved from public repositories. Subsequently, the obtained 946 *B. longum* subsp. *longum* chromosomes were processed through the DRep[85] and checkM[86] tools to cluster essentially identical genomes (ANI values = 99%) and select high-quality reference genomes from each replicate set. As a result, we obtained a non-redundant database of 277 *B. longum* subsp. *longum* strains, which was used for processing metagenomic fastq with inStrain tool under default parameters[33]. We used the value obtained by normalizing the genome coverage on the corresponding genome length as a proxy of the relative abundance of each *B. longum* subsp. *longum* reference strains.

## qRT-PCR analyses

To validate the highlighted strain dynamics of the infant gut microbiome, we performed a qRT-PCR-based assays focused on three cases in which a marked switch of the dominant genetic variant over time was observed.

Firstly, strain-specific primers were designed to target unique genetic sequences of the predicted persistent strains (Supplementary Data 6). The strain specificity of each designed primer pair was assessed in silico through primer-BLAST and in vitro validated by end-point PCR reactions performed by using the following thermal cycling protocol: 5 min at 94 °C for one cycle, followed by 30 cycles at 94 °C for 30 s, primer pair-specific annealing temperature for 30 s and 72 °C for 50 s, and a final cycle of 72 °C for 5 min. For this validation step, each PCR reaction was performed on the DNA extracted from each sample of the longitudinally collected fecal samples from a specific infant, together with DNA extracted from other taxa belonging to the same species of the targeted strain. Once the primer strain-specificity had been confirmed, a second end-point PCR was carried out using the DNA extracted from the fecal sample showing the higher average relative abundance of the strain containing the unique genetic sequence target of the designed primers. The obtained amplicon was then purified using the NucleoSpin PCR & Gel Clean Up kit (Macherey-Nagel, France), following the manufacturer's guidelines. The purified amplicon was subsequently used in a qRT-PCR run as the standard DNA to build a standard curve since no chromosomal DNA was available. Specifically, each qRT-PCR reaction mix contained 7.5 µl 2x SensiFast Sybr No-Rox kit (Meridian Bioscience, USA), 5 µl of DNA diluted to 10 ng/µl, each of the forward and reverse primer at 0.5 µM and nuclease-free water was added to obtain a final volume of 15 µl. Each qRT-PCR run was carried out on a CFX96 system (BioRad, CA, USA)

using the following protocol: 95 °C for 2 min, followed by 40 cycles of 95 °C for 5 s and 60 °C for 30 s, and a melting curve from 65 °C to 95 °C with increments of 0.5 °C/s. Negative controls (no DNA) for each primer set were included in each run, while standard curves were built using the CFX96 software (Biorad).

## Glycobiome prediction

The metagenomically assembled bifidobacterial genomes predicted to be either persistent or transient in the infant gut were screened for genes encoding the catalytic hydrolysis of the glycosidic bond. Prediction of glycosyl hydrolase (GH)-encoding genes and their classification into GH families were achieved through similarity search in the carbohydrate-active enzyme (CAZy) database[87] (BLAST cutoff e-value of $1 \times 10^{-10}$).

## Comparative genomic analyses

Only complete genome sequences belonging to *B. bifidum*, *B. longum*, *B. breve*, and *B. pseudocatenulatum* were retrieved from NCBI's RefSeq genome database and subjected to core-genome analysis using the Pangenome Analysis Pipeline (PGAP) v1.1 (--identity 0.5 --coverage 0.8 --cluster --method GF)[88]. Specifically, the predicted proteome of each bifidobacterial genome was screened for orthologues against the proteome of the other conspecific strains by BLAST analysis (cutoff e-value $< 1 \times 10^{-5}$ and 60% identity over at least 80% of both protein sequences). The resulting data were cataloged into functional gene clusters, also designated as Cluster of Orthologous Groups (COGs), employing MCL (graph-theory-based Markov clustering algorithm)[89], through the gene family (GF) method (cutoff e-value of $1 \times 10^{-10}$). A pan-genome profile was built using the algorithm provided as part of the PGAP software, based on a presence/absence matrix encompassing all COGs identified in the considered genomes. Accordingly, the protein families shared between *B. bifidum* and *B. longum* and absent in the *B. breve* and *B. pseudocatenulatum* genomes were collected.

## Genetic characterization of the shared *B. bifidum* and *B. longum* genetic traits and evaluation of their occurrence in infant, adult, and elderly populations

Protein sequences of the identified 14 COGs shared by *B. longum* and *B. bifidum* were subjected to extensive homology searches, and domain and localization predictions. In detail, Pfam v34.0 (https://pfam.xfam.org/), InterPro 86.0 (https://www.ebi.ac.uk/interpro/), and the Simple Modular Architecture Research Tool (SMART) (http://smart.embl-heidelberg.de/)[90] were employed to identify the protein domains, while SignalP v5, Psort v3.0.3, and THMM v2 were used for cellular localization prediction. SWISS-MODEL (https://swissmodel.expasy.org/) online tool was for 3D protein structure models and comparative modeling[91]. BLASTP and tBLASTN search with an E-value of 1e$^{-5}$ was performed against the integrated non-redundant protein sequence data resources (nr) for functional annotation of the coding sequences. BLASTP analysis (E value cutoff of 1e$^{-5}$) was also used to screen for PDC homologs in the *B. bifidum* and *B. longum* genomes identified in the multi-population metagenomes. Moreover, each of the 14 identified PDCs was aligned with the WGS reads to determine their prevalence and abundance in the population cohort, as previously described[92]. Briefly, following quality filtering (minimum mean quality score, 20; window size, 5 bp; quality threshold, 25; and minimum length, 100 bp) and removal of reads that map on the *Homo sapiens* genome, the final mapping against the 14 PDCs was performed using Bowtie2[93] through multiple-hit mapping and "very-sensitive" policy. The mapping was performed using a minimum score threshold function (−score-min C, −13,0) to limit reads of arbitrary length to two mismatches and retain those matches with at least 99% full-length identity. HTSeq software[94] (running in union mode) was employed to calculate read counts corresponding to each PDC gene.

## Analysis of fecal metagenomic samples of 132 mother-infant pairs

A total of 164 metagenomic fecal samples from mothers and their healthy term vaginally derived newborns were retrieved from public repositories (PRJEB6456, PRJNA475246). Specifically, we selected DNA sequencing data generated from shotgun metagenomic sequencing using Illumina platforms. For identification of vertical transmission events involving *B. longum* subsp. *longum* strains, we used StrainGE tool on fecal samples from the mother at delivery and newborn at 14–30 days after birth as described above. In addition, Bowtie2 was employed to map metagenomic reads against the sequence of GH136, and HTseq software was used to compute reads row counts as described above.

## Growth of *B. bifidum* PRL2010 on mucin and RNA-Seq analyses

*B. bifidum* PRL2010 cells were grown at 37 °C under anaerobic conditions (2.99% H2, 17.01% CO2, and 80% N2) (Concept 400; Ruskin) in De Man-Rogosa-Sharpe (MRS) broth (Sharlau Chemie, Barcelona, Spain) supplemented with 0.05% (wt/vol) l-cysteine hydrochloride. Viable cells were inoculated in 30 ml of freshly prepared modified MRS without glucose (mMRS) supplemented with 0.5% mucin. Cells were inoculated with an OD$_{600nm}$ of 0.1. After inoculum, growth was monitored, and at an OD$_{600nm}$ between 0.6 and 0.8 (exponential phase), cells were harvested by centrifugation at 6000 rpm for 5 min. Growth assays were carried out in triplicate. Total RNA of *B. bifidum* PRL2010 cultures was isolated as previously described[27]. The quality of the RNA was verified by employing a Tape station 2200 (Agilent Technologies, USA). RNA concentration and purity were evaluated using a spectrophotometer (Eppendorf, Germany). For RNA sequencing (RNA-Seq), from 100 ng to 1 μg of extracted RNA was treated to remove rRNA using the QIAseq FastSelect – 5 S/16 S/23 S following the manufacturer's instructions (Qiagen, Germany). The yield of rRNA depletion was checked using a Tape station 2200 (Agilent Technologies, USA). Subsequently, a whole transcriptome library was constructed using the TruSeq Standard mRNA Sample preparation kit (Illumina, San Diego, USA). Samples were loaded into a NextSeq high-output v2.5 kit (150 cycles, single end) (Illumina) following the technical support guide. Demultiplexed reads were quality filtered (with overall quality and length filters) and aligned to the *B. bifidum* PRL2010 reference genome through BWA[95]. Counts of reads that overlap ORFs were performed using HTSeq software[94]. Analysis of the RPKM values and false discovery rate correction (cut-off 0.01) was performed using DESeq2[96] and the formula RPKM = numReads/(geneLength/1000 × totalNumReads/1,000,000)[97]. The experiment was conducted in triplicate.

## Human cell line trials

Caco-2 cells, derived from a colorectal adenocarcinoma of a human male donor (purchased from ATCC) and HT29-MTX, a human colon carcinoma-derived, mucin-secreting goblet cell line from a female donor (kindly provided by prof. Antonietta Baldi, University of Milan) were cultured in Minimum Essential Medium (MEM) and Dulbecco's Modified Eagle's medium (DMEM) with high glucose (4.5 g/L) and 10 mM of sodium pyruvate, respectively, as previously described[98]. Both media were supplemented with 10% Fetal Bovine Serum (FBS), 2 mM glutamine, 100 μg/ml streptomycin, and 100 U/ml penicillin. Cultures were maintained at 37 °C in a 5% CO2 humidified atmosphere in 10-cm dishes and passaged three times a week. Subsequently, a mixed suspension of Caco-2 and HT29-MTX cells was seeded in DMEM + FBS at a density of ≈10$^5$ cells/cm$^2$ into cell culture inserts with membrane filters (pore size 0.4 μm) for Falcon 24-well-multitrays (Becton, Dickinson & Company, Franklin Lakes, NJ, USA). Cells were grown for 21 days until a tight monolayer was formed (TEER > 600 Ω cm$^2$) with a medium replacement every three days.

## Co-cultures of human cell monolayers and bifidobacteria

After 21 days from seeding, the culture medium of the 24-well plates was replaced with fresh, antibiotic-free DMEM. Subsequently, bifidobacterial cells with a final concentration of ≈$10^8$ cells/ml were inoculated on the Caco-2/HT29-MTX cell monolayers, as previously described[99]. The 24-well plates were then incubated at 5% $CO_2$ at 37 °C. After 4 h of incubation, bacterial cells were recovered in RNA later and stored at −80 °C until processing.

For these experiments, *B. longum* subsp. *longum* PRL2022 (harboring the GH136-encoding gene) and *B. longum* subsp. *longum* 1898B (lacking the GH136-encoding gene) were grown in MRS broth in anaerobic conditions at 37 °C. Once the exponential growth phase (0.6 <$OD_{600nm}$< 0.8) was reached, bifidobacterial cells were enumerated by using the Thoma cell counting chamber (Herka), diluted to reach a final concentration of $10^8$ cells/ml, washed in PBS, resuspended in 400 µl of antibiotic-free DMEM, and seeded on Caco-2/HT29-MTX cell monolayers. Furthermore, bifidobacterial strains resuspended in DMEM and maintained at the same incubation conditions of the 24-well plates without any contact with human cell lines were used as sample control. All experiments were carried out in triplicate.

In addition, to test whether the hormonal fraction of the serum-supplement FBS did affect the expression of the GH136 gene, *B. longum* subsp. *longum* PRL2022 was differentially grown on the culture medium DMEM with and without FBS, following the same protocol described above. Subsequently, the obtained RNA was used to perform qRT-PCR experiments to assess any differences in the expression of GH136 between the two considered conditions (DMEM + FBS or DMEM without FBS). Specifically, reverse transcription to cDNA was performed with the iScript Select cDNA synthesis kit (Bio-Rad Laboratories, USA) using the following thermal cycle: 5 min at 25 °C, 20 min at 46 °C, and 1 min at 95 °C. The mRNA expression levels were assessed with SYBR green technology in qRT-PCR using the Power Up SYBR Green Mastermix (ThermoFisher Scientific, USA) on a Bio-Rad CFX96 system according to the manufacturer's instructions. For this purpose, rpoB-fw (CACGATGGTGCTGCGACCTTCCC), rpoB-rv (GACC TGACGGATACGACGGTTGCC), atpD-fw (CGTATGCCTTCCGCCGTGG GTTAC), atpD-rv (ACGTAGATGGCTTGCAGCGAGGTG), ldh-fw (GTGA TGGGCGAGCATGGCGACTC), ldh-rv (GGAGGCGAAGCGGTCTTGG TTGTC) were used as primers for the amplification of the housekeeping genes *rpoB*, *atpD*, and *ldh*, while primer pair GH136-fw (AGCG TCTCGAAGCACATCAA) and GH136-rv (AGATCATCAGCGAGGCGAAG) was used to quantify GH136 gene expression. The PCR was carried out according to the following cycle: initial hold at 95 °C.

## Eukaryotic RNA-Seq data analysis

Total RNA was extracted from human cell monolayers with RNeasy Mini Kit (Qiagen). All samples had an RNA integrity number (RIN) ≥ 8. For RNA sequencing (RNA-Seq), TruSeq Standard mRNA Sample preparation kit (Illumina, San Diego, USA) was used to prepare stranded cDNA libraries with poly dT enrichment from 0.1 µg to 4 µg of RNA extracted from each sample according to the manufacturer's instructions. The quality and quantity of each cDNA sample was assessed by a Tape station 2200 (Agilent Technologies, USA) and Qubit Fluorometer (Thermofisher). Subsequently, the cDNA libraries were sequenced using an Illumina NextSeq 500 high-output v2.5 kit (150 cycles, single end) (Illumina) according to the technical support guide. The fastq-MCF program was used for trimming RNA-Seq raw data (fastq) based on quality score and the presence of adapter sequence. High-quality fastq were aligned to the Human reference genome sequence (GRCh38.p13) by using the splice-aware STAR algorithm (version 2.7.10a)[100], and the quality of alignments was evaluated using Picard software tool (version 2.26.11) (https://broadinstitute.github.io/picard/). Subsequently, quantification of reads mapped to individual gene transcripts was achieved through htseq-counts script of HTSeq software in "union" mode[94]. Raw counts were then normalized using CPM (Counts per million mapped reads) for filtering genes with low counts (CPM < 1) and TMM (Trimmed Mean of M-Values) for statistically robust differential gene expression analysis through the EdgeR package[101]. The expression difference was evaluated as log2 fold change (logFC) of average expression in each sample pair of compared groups (co-cultures Caco-2/HT29-MTX/B. longum subsp *longum* 1898B and Caco-2/HT29-MTX/*B. longum* subsp. *longum* PRL2022). In addition, a Volcano plot was created for each comparison to simultaneously visualize expression changes (log fold change) and their statistical significance (*p*-value).

## *B. longum* subsp. *longum* RNA-Seq data analysis

Extraction of total RNA from *B. longum* subsp. *longum* strains, RNA sequencing, and raw fastq processing were performed as described above for the *B. bifidum* RNA-sequencing experiment. Generation of raw counts and identification of DEGs were achieved as described above for the eukaryotic RNA-seq data analysis.

## Human retrospective clinical studies

Shotgun metagenomic datasets of fecal samples were retrieved from publicly available human clinical trials in which female and male subjects consumed daily doses of microbial formulations containing *B. longum* subsp. *longum* strains (PRJNA324129, PRJEB28097). The chromosomes of the bacterial strains used in the selected studies (ANI value of 98.5%) were manually inspected to assess the presence/absence of the GH136 gene, resulting in two different treatment patterns. A group of 21 healthy volunteers (11 females and 10 males) received *B. longum* subsp. *longum* AH1206, which was found to possess the GH136 gene (daily dose of $10^{10}$ viable cells for a 4-week period). For this latter cohort, fecal samples were collected at the baseline period, after 28 days of treatment, and about 200 days after consumption cessation. A second group of healthy individuals ($n = 10$; 4 females and 6 males) consumed a probiotic supplement ($5 \times 10^9$ CFU bi-daily for a 4-week period) containing *B. longum* subsp. *longum* devoid of the GH136 gene (named strain AG1). Corresponding fecal samples were analyzed at baseline, on day 21 of microbial intervention, and after 28 days of follow-up. Genome sequence of the strain AH1206 was acquired from the NCBI database, while the chromosome of AG1 was recovered from the publicly available metagenomic sequenced reads of the used probiotic supplement using Spades v3.15 (metagenomic mode) with default parameters. Taxonomic classification of the assembled contigs was achieved using METAnnotatorX2 pipeline with manually curated genome databases. Completeness and contamination of reconstructed chromosome of *B. longum* subsp. *longum* AG1 was validated through CheckM analysis[86]. Test of *B. longum* subsp. *longum* persistence was performed through reads mapping with Bowtie2, as described above. Briefly, metagenomic reads were filtered to remove low-quality (score lower than 20), tRNA, and rRNA sequences. Moreover, the sequences mapping against *Homo sapiens* genome were also eliminated. Filtered reads were used as input for Bowtie2 (--very-sensitive option, with at least 99% full-length identity). HTSeq software (running in union mode) was used to calculate read counts corresponding to each reference genome[94]. The relative abundance of mapped reads was estimated for each metagenomic sample by normalizing the raw reads count on the total number of sequenced reads.

## Statistics and reproducibility

To guarantee high resolution and consistency of the input data, we selected shotgun metagenomics data sets only based on the Illumina sequencing platform. Moreover, samples from individuals with reported intestinal morbidity were excluded a priori. The sample size was mainly determined by the availability of resources. Therefore, no statistical method was used to predetermine sample size. The experiments were not randomized. The investigators were not blinded to allocation during experiments and outcome assessment, as this was an observational study with no randomization used. The software SPSS version 25, and ORIGIN version 9.8.0.200 (www.ibm.com/software/it/

analytics/spss/) (https://www.originlab.com/) were used for statistical data analyses and graphing. PERMANOVA was calculated using the adonis2 function from the vegan R package. PERMANOVA analyses based on Bray-Curtis measures of species-level abundance data were conducted using 1000 permutations to estimate *p*-values for the observed differences between the compared groups in PCoA analyses. In differential gene expression analysis, EdgeR package was used to estimate the statistical significance of differences between fold changes as the False Discovery Rate (FDR).

### Reporting summary
Further information on research design is available in the Nature Portfolio Reporting Summary linked to this article.

## Data availability
The infant shotgun metagenomic sequencing and RNA-seq data of human intestinal cells, *B. longum* subsp. *longum* PRL2022 and 1898B generated in this study have been deposited in the Sequence Read Archive (SRA) under accession code PRJNA833139. All other publicly available metagenomic datasets supporting the findings of this study can be obtained through the accession code reported in the Supplementary Data. The Human reference genome sequence (GRCh38.p13) used in this study is available from https://www.ensembl.org/Homo_sapiens/Info/Index?db=core. The genome sequence of *B. longum* subsp. *longum* 1898B and PRL2022 are accessible through https://www.ncbi.nlm.nih.gov/assembly/GCA_002075875.1 and https://www.ncbi.nlm.nih.gov/assembly/GCA_016759745.1, respectively. Carbohydrate-active enzyme (CAZy) database is obtainable from http://www.cazy.org/.

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

## Acknowledgements
We thank GenProbio Srl for the financial support of the Laboratory of Probiogenomics. Part of this research is conducted using the High-Performance Computing (HPC) facility of the University of Parma. This research has financially been supported by the Programme "FIL-Quota Incentivante" of University of Parma and co-sponsored by Fondazione Cariparma". D.v.S. is a member of APC Microbiome Ireland funded by Science Foundation Ireland (SFI), through the Irish Government's National Development Plan (Grant no. SFI/12/RC/2273-P1 and SFI/12/RC/2273-P2). G.T. has been supported by "Fondazione Cariparma" in the framework of the project entitled "Parma Microbiota". LMV has been supported by by "Programma Operativo Nazionale 2014–2020 of the Italian Ministry of University and Research. The funding from Project AGL2017-83653R (Spanish "Ministerio de Ciencia, Innovación y Universidades (MCIU)", "Agencia Estatal de Investigación (AEI)" and FEDER) is also acknowledged.

## Author contributions
Conceptualization, supervising, coordination, and revising of manuscript by F.T., O.V., D.v.S., C.M., and M.V. RNAseq and bacterial DNA sequencing experiments by G.A., G.L., R.A., and A.V. Collection data and genomic, metagenomic, and transcriptomic data processing and analysis by C.T., G.A., S.M.R., M.B., C.A., L.M.V., S.A., G.T., M.C., and M.G. Assisting in computational framework development, implementation, and computational analyses by F.F., G.A.L., and L.M. Drafting of manuscript by C.T., G.A., and C.M. All authors critically revised and approved the manuscript.

## Competing interests
The authors declare no competing interests.
