## [Peer Review File · Nature Communications]

REVIEWER COMMENTS

Reviewer #1 (Remarks to the Author):

Tarrachini et al. present an interesting analysis of the persistence of gut microbes and the findings warrant attention. However, to ensure sufficient level of robustness, it would be necessary to perform some additional validity analyses since the numbers of samples are fairly limited. Firstly, the authors have missed one of the most relevant papers for their application, by Shao et al. Nature, DOI: 10.1038/s41586-019-1560-1. The study presents a cohort of neonatal samples from babies born with either vaginal or c-section delivery, and includes also samples from a later time point during the first year and samples from the mothers as well. The large size and high sequencing depth of this study makes it ideal for both validity purposes and for improving the analysis of the target species.

The approach taken by the authors has a number of concerns of technical nature. Firstly, the sequencing depth of the study cohort is not impressive, and the authors choose to even cut it further down to 5M reads to normalize the depth per sample. This is not an optimal approach as it loses valuable information, instead the authors should use all the available reads and make a post hoc correction to the analysis, should it become evident that the conclusions are correlated with the depth. It could well be that the depth variation shows no systematic correlation with the ability to identify strains of the target species. Second, the way the authors choose to construct MAGs and use those with published genomes in the StrainGE method makes it uncertain how well the identified strains represent the variation actually present in the metagenomic samples. Standard MAGs are notoriously noisy and maybe seriously convoluted, which limits their use for the study purposes. To ensure that the conclusions are not biased by the method, the authors could use a more recent high-resolution metagenomic profiling and phylogenetic analysis approach taken by Mäklin et al. DOI: 10.1038/s41467-022-35178-5 to re-analyze the Shao et al. neonatal cohort data. By providing a more refined reference set of genomes of the target species than the NCBI RefSeq database used in StrainGE, together with an adequate biological subgrouping of them, the authors may gain considerable additional resolution to make sure if a strain truly persists in a host and even see additional within-host variation concerning the target species. Again, the Shao et al. data could be used both to do a comparison of the persistence pattern with the study cohort and also to construct an additional set of reference genome assemblies for the analysis pipeline employed by Mäklin et al.

Reviewer #2 (Remarks to the Author):

The authors of this article analyzed a large collection of analyses of metagenome data on longitudinal infant data, which also included public data sets, comparative genome analyses, wet lab experiments and clinical trials to corroborate the findings. Methodology is sound and authors provide enough details on pipelines and software used. Authors provided all the details on where to find the data used in public repositories.

Authors showed a preferential strain-level persistence across life stages that is sex specific, corroborated by the additional experiments and the data. Certain stains are persistent in females, probably due to sex hormones, although the direct impact of sex hormones have not been tested.

The results showed that specific bacterial glycosyl hydrolases related to host-glycan metabolism may contribute to this more efficient colonization in females compared to males.

My comments to authors:

Q1. The authors suggest present the hypothesis of sex-hormones, but could they elaborate more on the potential links between sex hormones and bacterial glycosyl hydrolases?

And are there other mechanisms that could be involved in regulation of host-glycan metabolism that could be sex specific?

Q2. Page 12 – line 231. It would like to see the number of males and females in each age category. I found confusing that on the text authors mention both children (5-18) group and adults (19-55), while in Suppl Table 7 I can only see the categories: infant, adults, elderlies. Is the adult category in suppl table 7 comprising children? If so, I would like to see the statistics when children and adults are separated. This is essential to support the statement that persistence in females is higher at reproductive age

Q3. Page 13 – in the difference between males and females, would author expect/could account for potential confounders such as dairy food consumption/lactase persistence? Probably this type of information is not available on public data sets but geographic regions could be used as potential proxies (like south / north Europe for lactase persistence and Europe vs China for dairy food consumption). It would be intriguing to see the prevalence of *B. longum* species and *B. bifidum* species by geographic groups.

We have addressed all comments made by the two referees, and we have implemented the vast majority of the proposed changes. All the comments were considered very helpful and valuable and have helped to significantly improve the revised version of the manuscript in its overall quality. The final draft of our manuscript, in our opinion, is now of much better quality.

In the following, we will discuss the individual suggestions/comments made by the reviewers in detail:

Reviewer #1 (Remarks to the Author):

Tarrachini et al. present an interesting analysis of the persistence of gut microbes and the findings warrant attention. However, to ensure sufficient level of robustness, it would be necessary to perform some additional validity analyses since the numbers of samples are fairly limited. Firstly, the authors have missed one of the most relevant papers for their application, by Shao et al. Nature, DOI: 10.1038/s41586-019-1560-1. The study presents a cohort of neonatal samples from babies born with either vaginal or c-section delivery, and includes also samples from a later time point during the first year and samples from the mothers as well. The large size and high sequencing depth of this study makes it ideal for both validity purposes and for improving the analysis of the target species.

The approach taken by the authors has a number of concerns of technical nature. Firstly, the sequencing depth of the study cohort is not impressive, and the authors choose to even cut it further down to 5M reads to normalize the depth per sample. This is not an optimal approach as it loses valuable information, instead the authors should use all the available reads and make a post hoc correction to the analysis, should it become evident that the conclusions are correlated with the depth. It could well be that the depth variation shows no systematic correlation with the ability to identify strains of the target species. Second, the way the authors choose to construct MAGs and use those with published genomes in the StrainGE method makes it uncertain how well the identified strains represent the variation actually present in the metagenomic samples. Standard MAGs are notoriously noisy and maybe seriously convoluted, which limits their use for the study purposes. To ensure that the conclusions are not biased by the method, the authors could use a more recent high-resolution metagenomic profiling and phylogenetic analysis approach taken by Mäklin et al. DOI: 10.1038/s41467-022-35178-5 to re-analyze the Shao et al. neonatal cohort data. By providing a more refined reference set of genomes of the target species than the NCBI RefSeq database used in StrainGE, together with an adequate biological subgrouping of them, the authors may gain considerable additional resolution to make sure if a strain truly persists in a host and even see additional within-host variation concerning the target species. Again, the Shao et al. data could be used both to do a comparison of the persistence pattern with the study cohort and also to construct an additional set of reference genome assemblies for the analysis pipeline employed by Mäklin et al.

REPLY: We thank the reviewer for the constructive suggestions, which have been implemented in the revised version of the manuscript to the best of our ability. In detail, the method for controlling sequencing depth by subsampling the dataset to 5M reads was applied to the public

datasets, while our study cohort encompassing the 11 infants was parsed using all available reads. In this latter case, the sequencing depth was controlled by normalizing the number of persistent strains in each sample (detected both in pre-weaning and at 12 or 24 months) using the ratio between the mean sequenced reads across the whole dataset and the number of reads obtained from a given sample. With this method, the number of strains detected as persistent is now proportional to the sequencing depth and thus comparable across samples.

We have detailed this procedure in the methods section (page 27, lines 493-499), and we have revised the main text accordingly (page 6, lines 121-128).

In general, we agree with this referee regarding the possibility of losing some information by rarefying the data. For this reason, exploiting a subset of the infant cohort from Shao et al., we compared the strain-level profiles of the target bifidobacterial species obtained using all available reads and a cut-off number of 5,000,000. After normalizing the results obtained in the dataset encompassing all available reads, the comparison revealed some minor differences, mainly concerning the coverage values of the detected genomes. However, these discrepancies are not really relevant to our study, which was aimed at identifying the presence/absence of reference strains in metagenomic reads. In fact, the Mann-Whitney U test calculated on the strain patterns of each bifidobacterial species generated non-significant p-values, underscoring the ability to identify strains in a comparable measure in both cases, although only a small number of additional genomes were detected using all sequenced reads. We have integrated this information in Supplementary Table S8 and the main text (page 28, lines 534-540).

Moreover, the Shao et al. cohort was also employed both to construct valuable reference genomes of the target species and as a validation cohort to corroborate the findings observed in our study cohort. More specifically, as suggested by this referee, a new set of reference genomes of each target species was constructed using the pipeline proposed by Mäklin et al., which was also employed to validate strain persistence patterns of the bifidobacterial species across the infant weaning stage. We have incorporated this analysis in the materials and methods section (page 27, lines 500-515) and the main text (Page 6, lines 137-151).

Reviewer #2 (Remarks to the Author):

The authors of this article analyzed a large collection of analyses of metagenome data on longitudinal infant data, which also included public data sets, comparative genome analyses, wet lab experiments and clinical trials to corroborate the findings. Methodology is sound and authors provide enough details on pipelines and software used. Authors provided all the details on where to find the data used in public repositories.

Authors showed a preferential strain-level persistence across life stages that is sex specific, corroborated by the additional experiments and the data. Certain stains are persistent in females, probably due to sex hormones, although the direct impact of sex hormones have not been tested. The results showed that specific bacterial glycosyl hydrolases related to host-glycan metabolism may contribute to this more efficient colonization in females compared to males.

My comments to authors:

Q1. The authors suggest present the hypothesis of sex-hormones, but could they elaborate more on the potential links between sex hormones and bacterial glycosyl hydrolases?

And are there other mechanisms that could be involved in regulation of host-glycan metabolism that could be sex specific?

REPLY: We agree with this reviewer's comment. Accordingly, as suggested by this referee, in the revised version of the manuscript, we have extended the discussion concerning the possible role of host sex hormones on gender-specific differences in the gut microbiota composition (Page 14, lines 296-307). It should be considered that although growing evidence supports the important role of host sex hormones in shaping the gut microbiota, little is currently known about the mechanisms underlying this relationship. In this context, this study supports progress of our understanding on this phenomenon.

Q2. Page 12 – line 231. It would like to see the number of males and females in each age category. I found confusing that on the text authors mention both children (5-18) group and adults (19-55), while in Suppl Table 7 I can only see the categories: infant, adults, elderlies. Is the adult category in suppl table 7 comprising children? If so, I would like to see the statistics when children and adults are separated. This is essential to support the statement that persistence in females is higher at reproductive age

REPLY: We thank the reviewer for pointing out this unwanted omission. We have now reported in Supplementary Table 7 the prevalence of *B. bifidum*, *B. longum* subsp. *longum* in children and adults separately. Moreover, we have specified the number of females and males in each age group (page 13, lines 253-256).

Q3. Page 13 – in the difference between males and females, would author expect/could account for potential confounders such as dairy food consumption/lactase persistence? Probably this type of information is not available on public data sets but geographic regions could be used as potential proxies (like south / north Europe for lactase persistence and Europe vs China for dairy food consumption). It would be intriguing to see the prevalence of *B. longum* species and *B. bifidum* species by geographic groups.

REPLY: We thank this reviewer for the suggestion to consider dairy food consumption/lactase persistence as potential confounders. We also appreciate that the reviewer acknowledges the difficulty in finding this (meta)data for metagenomics datasets in public repositories. Accordingly, we followed the reviewer's indication to consider geographic regions as an indirect measure of dairy food consumption/lactase persistence, and we performed a new strain-level analysis to investigate the sex-dependent prevalence of the target species across geographic groups. To maintain data consistency, we employed the same methodology and genome databases used in our previous analyses. The main text and the materials and methods section have consequently been revised (page 13, lines 269-276; page 29, lines 545-552).

REVIEWERS' COMMENTS

Reviewer #1 (Remarks to the Author):

The authors have appropriately taken all comments into account, I have no further remarks apart from that a typo on p 6 should be corrected (thusn->thus).

Reviewer #2 (Remarks to the Author):

The authors have responded to my comments and I have not further requests.